# Vaccination against SARS-CoV-2 in Patients with Inflammatory Bowel Diseases: Where Do We Stand?

**DOI:** 10.3390/life11111220

**Published:** 2021-11-11

**Authors:** Phil-Robin Tepasse, Richard Vollenberg, Tobias Max Nowacki

**Affiliations:** Department of Medicine B for Gastroenterology, Hepatology, Endocrinology and Clinical Infectiology, University Hospital Muenster, 48149 Muenster, Germany; richard.vollenberg@ukmuenster.de (R.V.); tobias.nowacki@ukmuenster.de (T.M.N.)

**Keywords:** IBD, Crohn’s disease, ulcerative colitis, vaccination, SARS-CoV-2, COVID-19

## Abstract

Crohn’s disease and ulcerative colitis are chronic inflammatory bowel diseases (IBDs). Immunosuppressive medication is the main therapeutic approach to reducing inflammation of the gastrointestinal tract. Immunocompromised patients are more vulnerable to severe courses of illness after infection with common pathogens. The severe acute respiratory syndrome coronavirus-2 (SARS-CoV-2) is the pathogen of the coronavirus disease 2019 (COVID-19) pandemic. COVID-19 leads to acute respiratory distress syndrome (ARDS) following severe pulmonal damage in a significant number of cases. The worldwide circulation of SARS-CoV-2 has led to major concerns about the management of IBD patients during the pandemic, as these patients are expected to be at greater risk of complications because of their underlying altered immunological condition and immunosuppressive therapies. Vaccination against SARS-CoV-2 is considered the main approach in containing the pandemic. Today, several vaccines have been shown to be highly effective in the prevention of SARS-CoV-2 infection and severe disease course in subjects without underlying conditions in respective registration studies. Patients with underlying conditions such as IBD and/or immunosuppressive therapies were not included in the registration studies, so little is known about effectiveness and safety of SARS-CoV-2 vaccination in immunocompromised IBD patients. This review provides an overview of the recent knowledge about vaccine response in IBD patients after vaccination against SARS-CoV-2.

## 1. Introduction

Crohn’s disease (CD) and ulcerative colitis (UC) are chronic remittent inflammatory bowel diseases (IBDs). Immunosuppressive medication is the predominant therapeutic approach for reducing inflammation of the gastrointestinal tract [1,2]. Immunosuppressive therapies can lead to severe side effects, such as opportunistic infections, as well as susceptibility to severe illness after infection with a common pathogen [3].

The severe acute respiratory syndrome coronovirus-2 (SARS-CoV-2) is the pathogen of the coronavirus disease 2019 (COVID-19) pandemic [4]. In a significant number of cases, COVID-19 leads to acute respiratory distress syndrome (ARDS) following severe pulmonal damage [5]. Hyperinflammation following infection with SARS-CoV-2 is considered a relevant mechanism leading to progressive lung failure and, in some cases, to multiorgan failure and death [6,7]. The pandemic has generated major concerns about the management of IBD patients, as recent studies have suggested more severe courses of disease in IBD patients due to their underlying altered immunological condition and immunosuppressive therapies [8].

Until recently, only a few therapies were shown to be effective in combatting COVID-19. Glucocorticoid dexamethasone and the interleukin-6 antagonist tocilizumab were shown to significantly reduce hyperinflammation and mortality in severely and critically ill COVID-19 patients [9,10,11]. Despite these therapies, mortality is still high, especially in older patients or patients with known medical conditions such as chronic heart, renal, or lung disease and in patients undergoing immunosuppressive or cancer therapies [12,13].

Vaccination against SARS-CoV-2 is considered the most suitable approach for containing the pandemic. Several vaccines have been approved for application in humans and have been shown to be highly effective in the prevention of SARS-CoV-2 infection and severe disease course in subjects without underlying conditions in respective registration studies. Patients with underlying conditions such as IBD, or those undertaking immunosuppressive therapies, were not included in the registration studies. As a result, little is known about the effectiveness, side-effects, and safety of SARS-CoV-2 vaccination in IBD patients, especially in the case of required immunosuppressive therapies. Recent studies revealed attenuated immune responses to SARS-CoV-2 infection in patients treated with immunomodulating therapies (e.g., anti-TNF agents), suggesting an attenuated immune response following vaccination [14].

This review offers an overview about current knowledge regarding vaccine responses in IBD patients after vaccination against SARS-CoV-2.

## 2. Review of Literature

### 2.1. Methods

We performed an electronic database search using PubMed and searched for studies accepted to PubMed during 2020 until 30 August 2021. We conducted the PubMed search using the keywords “IBD”, “inflammatory bowel disease”, “COVID-19”, and “vaccination”. The keywords were combined using the operator “AND”. Three reviewers independently screened the titles of the studies retrieved from the search. Only studies addressing the effectiveness and safety of anti SARS-CoV-2 vaccines in IBD patients were selected for further reading. Of them, only studies investigating humoral and cellular response and safety were included in the review. Selected studies were screened for further available literature regarding safety and humoral/cellular response in IBD patients. To address recommendations of SARS-CoV-2 vaccination in IBD patients, international guideline articles were also included and screened for further literature addressing the above-mentioned criteria. The selected titles from each of the three reviewers were combined and duplicates were removed. The included articles were reviewed for IBD cohort characteristics, IBD medication, humoral responses (seroconversion rates, antibody levels), and cellular responses in the context of IBD medication and safety data after vaccination.

All selected studies revealed data on seroconversion rates with the following anti-SARS CoV-2 spike antibody concentration cut-off values while using different assays: Kennedy et al. ≥15 U/mL; Reuken et al. ≥33.8 BAU/mL and ≥13 AU/mL; Kappelmann et al. ≥1µg/mL; Wong et al. ≥10 sCOVG Index; Podzdyakova et al. ≥50 AU/mL. Reuken et al. and Kennedy et al. presented data after the first vaccination (BNT162b2 and ChAdOx1) and Reuken et al., Pozdnyakova et al., Spencer et al., Wong et al., Kappelmann et al., and Kennedy et al. presented data up to 90 days after second vaccination (BNT162b2, ChAdOx1, mRNA-1273, and Ad26.CoV2.S). To conduct a meta-analysis on seroconversion rates, we extracted complete data of IBD patients (healthy controls were not included) of each study (e.g., absolute numbers of study participants, absolute numbers of participants with and without seroconversion as defined by the authors, and IBD medication) and calculated the percentage proportion of seroconverted participants.

### 2.2. SARS-CoV-2 Vaccines

At the time of writing, four SARS-CoV-2 vaccines have been approved in Germany and most European countries. All four vaccines target the SARS-CoV-2 spike protein of the virus, which binds to the ACE receptor on human cells and is essential for the cell entry of the virus. The mRNA-1273 (Moderna/National Institute of Health) and BNT162b2 (BioNTech/Pfizer) vaccines are both mRNA vaccines. In both cases, the mRNA is packed into lipid nanoparticles, delivering the mRNA into human cells after injection. Cells are then able to synthesize the spike protein, which was shown to be able to induce both neutralizing antibodies and T-cellular immune responses. In phase III trials, both vaccines showed an efficiency above 94% in preventing COVID-19 in the study population, accompanying a good safety profile [15,16]. The Ad26.CoV2.S (Johnson & Johnson) and ChAdOx1 (Astrazeneca) vaccines are replication-inactivated adenovirus vector vaccines from a chimpanzee adenovirus that express the SARS-CoV-2 protein in full length after entry into the host cell. Both vaccines were shown to be able to induce neutralizing antibodies and strong T-cell responses. Phase III trials revealed an efficiency in the prevention of SARS-CoV-2 infection of approximately 70% [17,18].

### 2.3. Immunological Response to SARS-CoV-2 Vaccines in IBD Patients

The phase III trials of the above-mentioned vaccines did not include patients using immunosuppressive therapies, which is why little is known about safety and efficiency in these patients. Meanwhile, a rising number of studies found an impaired immune response in patients using immunosuppressive therapies compared to healthy controls, which has even led to the suggestion of applying a third dose of the vaccine to selected patients [19]. While quite a few studies on safety and efficiency in solid organ transplant recipients are available [20,21,22], only a small number of studies on IBD patients have been recently published. These studies revealed a varying immunological response in IBD patients depending on the type of vaccine and on immunosuppressive therapy. Kennedy et al. found an attenuated humoral response in patients treated with infliximab (n = 865) compared to vedolizumab (n = 428) after the first dose of either the BNT162b2 or the ChAdOx1 vaccine. Serum antibody concentrations were lower and seroconversion rates were less common. Smoking, Crohn’s disease, an age of over 60 years, and additional immunomodulator use were identified as independent risk factors for an attenuated immune response. Seroconversion rates after a single dose of either vaccine were higher after prior SARS-CoV-2 infection. A small number of participants (n = 27) without evidence of prior SARS-CoV-2 infection received a second vaccine dose of BNT162b2, which induced increasing antibody levels and a seroconversion rate in 85% (17/20, infliximab) and 86% (6/7, vedolizumab) of patients, suggesting a significantly improved immune response after a second vaccine dose [23]. Wong et al. found a serological conversion rate of 100% in 26 IBD patients, who completed two doses of vaccination with BNT162b2 or mRNA-1273 (TNF antagonist (n = 8), vedolizumab (n = 12), ustekinumab (n = 2), no medication (n = 4)). Both anti-TNF agents and vedolizumab were associated with lower antibody levels compared to healthy controls. No differences in the frequency of side effects were detected in comparison of both IBD and control groups [24]. Kappelmann et al. found no difference in seroconversion rates and antibody levels between vaccine groups (BNT162b2 and mRNA-1273) and IBD medication after two vaccine doses. In the overall study group, 300 of the 317 (94.6%) participants had detectable antibodies after two vaccine doses. The study population was divided into one group using corticosteroids (n = 13) and one group without corticosteroid use (n = 304), and a difference between seroconversion rates was found (84.6% vs. 95.1%). Taking this finding into account, the authors discussed a possible attenuating effect of corticosteroids on vaccine response [25]. Pozdnyakova et al. compared seroconversion rates in IBD patients with different immunosuppressive therapies after vaccination with the vector-based vaccine Ad26.CoV2.S (n = 12) and the mRNA vaccines BNT162b2 (n = 193) and mRNA-1273 (n = 148) and found a lower seroconversion rate two weeks after full (two doses) vaccination with Ad26.CoV2.S (90%) compared to vaccination with the mRNA BNT162b2 vaccine (99%) and mRNA-1273 vaccine (100%). Quantitative anti-Spike IgG two weeks and eight weeks after full vaccination was significantly higher after mRNA vaccination compared to vaccination with the vector vaccine. On the other hand, no difference in seroconversion rates was found between IBD therapy subgroups [26]. Spencer et al. found 100% seroconversion rates in children with IBD (n = 20), regardless of IBD therapies (95% of participants received biological therapies) [27]. Only one study investigating cellular immune responses is available. Reuken et al. found significant T-cellular immune responses after a first dose of vaccination with the BNT162b2 or ChAdOx1 vaccine in 27 of 28 immunocompromised IBD patients without differences in medication subgroups. A second vaccine dose maintained the cellular immune response and improved the seroconversion rate from 71.4% to 91.7% in IBD patients (versus 85.1% and 100% in healthy controls, statistically not significant). Antibody levels were slightly, but not significantly, lower in IBD patients compared to healthy controls [28] (Table 1).

### 2.4. Meta-Analysis of Seroconversion Rates in IBD Patients

First, we screened the selected literature for data demonstrating seroconversion rates in IBD patients (Figure 1).

Next, we conducted a meta-analysis of seroconversion rates as described in the methods section. For meta-analysis of seroconversion rates after first vaccination with the BNT162b2 or ChAdOx1 vaccine, the data of 895 participants were available. Of them, 363 (40.6%) participants seroconverted up to two weeks after vaccination. Data up to 90 days after second vaccination with BNT162b2, ChAdOx1, mRNA-1273, and Ad26.CoV2.S from 676 participants were available. Of them, 652 (96.4%) participants seroconverted after the second vaccination (Figure 2).

### 2.5. Effectiveness of SARS-CoV-2 Vaccination in IBD Patients

The effectiveness of vaccination against infection with SARS-CoV-2 has not been examined prospectively, but Khan et al. retrospectively analyzed 14.679 IBD patients in a veteran cohort, of whom at least 7321 received one dose of the vaccine. The overall cohort received a broad spectrum of immunosuppressive therapies (anti-TNF agents, vedolizumab, ustekinumab, steroids, immunomodulating therapies). Median age was 68 years and 61.8% had ulcerative colitis. In total, 45.2% received the Pfizer vaccine and 54.8% received the Moderna vaccine. This study found an effectiveness of 80.4% against SARS-CoV-2 infection seven days after two vaccinations compared to the unvaccinated cohort. No differences were found in IBD medication subgroups. Overall mortality was not significantly different in vaccinated and unvaccinated patients, perhaps due to relatively low numbers of infection in both groups [30].

### 2.6. Safety of SARS-CoV-2 Vaccination in IBD Patients

Only minimal data are available regarding the safety of SARS-CoV-2 vaccination in immunocompromised IBD patients. Botwin et al. longitudinally surveyed 246 IBD patients who received at least one dose of a two-dose vaccination series of either the BNT162b2 or the mRNA-1273 vaccine. Adverse events (AE) were documented for eight days following each dose. In total, 80% of patients received multiple IBD medications (anti-TNF agents, vedolizumab, ustekinumab, fofacitinib, corticosteroids) and 20% did not receive any immune-compromising therapies at the time of vaccination. Moreover, 57% received BNT162b2 and 42.7% received mRNA-1273, while 67% had Crohn’s disease and 33% ulcerative/indeterminate colitis. The rates of AE after the first and second vaccine doses were comparable to previously reported rates in healthy subjects, and severe AEs were considerably rare. AEs were more common in UC patients than in CD patients (78% vs. 55%). Younger age was identified as a risk factor for increased AE rates, whereas AEs in individuals receiving any immunomodulating therapy at the time of vaccination was less common. This is the largest available study in this field; taken together, these data suggest a safe profile for SARS-CoV-2 mRNA vaccines in IBD patients in relation to immunomodulating therapies during the first eight days following either one or two vaccine doses [29].

## 3. Conclusions

Available studies reveal promising data on vaccination effectiveness and safety in immunocompromised IBD patients, showing high seroconversion rates, strong T-cell responses and pronounced protection against SARS-CoV-2 infection despite ongoing immunosuppressive therapies without relevant side effects. While occasional case reports on influenza vaccinations have raised the concern that vaccinations could trigger a flare of the IBD, larger studies on H1N1 vaccinations in IBD have shown no effect on disease activity in vaccinated patients. In a study with >500 IBD patients, Rahier et al. showed that only 3.9% of participants experienced a self-limiting clinical disease flare after influenza vaccine administration [31], supporting the hypothesis that there is no increased risk of IBD re-exacerbation after vaccination. The current literature seems to suggest that this is probably also true for SARS-CoV-2 vaccinations. This is in line with findings in patients with rheumatic and musculoskeletal diseases (RMD). The overwhelming majority of patients tolerated their vaccination well, with rare reports of inflammatory RMD flare (5%; 1.2% severe) and only very few reports of severe adverse events (0.1%) [32].

While a flare up of underlying IBD following vaccination appears to be of minor concern, there is the possibility that patients taking immunosuppressive medication will have a reduced vaccine effectiveness due to impaired immune responses. Some studies highlight the importance of full vaccination, leading to significantly improved immune responses after the second vaccine dose of mRNA and vector vaccines [28]. The data indicate stronger immune responses after mRNA vaccination compared to vaccination with vector vaccines [26]. Most studies found no differences in immune responses between immunosuppressive regimens and revealed seroconversion rates and antibody levels comparable to healthy controls without immunosuppressive therapies (seroconversion rates > 90%). However, data on vector vaccine or even mRNA-based vaccine effectiveness in IBD patients under immunosuppression is scarce. Few data may indicate slightly attenuated humoral responses in patients receiving infliximab, corticosteroids, or immunomodulating therapies, especially in cases of combination therapies. Patients receiving anti-TNF monotherapy appear to have reduced protective immunity in response to hepatitis A and B as well as influenza and pneumococcal vaccinations [33,34,35]. Data on vaccine effectiveness against common pathogens such as influenza and hepatitis B under “newer” immunosuppressive medications such as ustekinomab or tofacitinib are missing in IBD patients, and only heterogeneous results with diminished protective immunity following pneumococcal vaccinations but preserved influenza immunity are reported in patients with rheumatoid arthritis [36].

In comparison, an increasing number of studies on SARS-CoV-2 vaccinations in solid organ transplant (SOT) recipients revealed a significantly attenuated immune response in these patients. For example, Mazzola et al. found a seroconversion rate of only 28.6% 28 days after the second SARS-CoV-2 vaccination with the BNT162b2 vaccine in SOT recipients receiving immunosuppressive therapies with calineurin inhibitors, mycophenolic acid, mTor inhibitors, and steroids [37]. Perry et al. demonstrated a severe impairment of humoral response after the second vaccination with the BNT162b2 vaccine in lymphoma patients actively treated with B-cell depleting therapies (rituximab, obinutuzumab), revealing a seroconversion rate of only 7.3% in these patients [38].

Many national and international regulatory and advisory boards have published vaccination recommendations aiming at sufficiently protecting patients at increased risk of contracting COVID-19 and experiencing a complicated or even fatal course of the disease. As, to date, there are no prospective studies reporting the effect of any SARS-CoV-2 vaccine specifically in patients with IBD, only few expert guidelines on SARS-CoV-2 vaccinations for IBD patients exist or were published in the early stages of the pandemic. A position statement by the British Society of Gastroenterology Inflammatory Bowel Disease section and IBD Clinical Research Group was issued in the beginning of 2021, at a time in which the British healthcare system was under considerable strain and vaccine availability was limited [39]. Meanwhile, vaccine availability has improved in many Western countries, while others still experience considerable vaccine shortages. Considering the above-reviewed actual literature demonstrating low risks and high benefits of SARS-CoV-2 vaccinations, we strongly support protective vaccinations for most IBD patients, which is in line with other expert opinions from early 2021 [39], since the risk of vaccination in this patient population is low and the main concern is suboptimal development of protective immunity, especially in patients under immunosuppressive therapy rather than side effects or flare of the underlying IBD.

Given the theoretical risk of suboptimal immune responses on the one hand and an apparently higher effectiveness of mRNA-based vaccines, there could be a rationale for a preferred use of these vaccines in IBD patients. Additionally, IBD patients are typically of a younger age and carry an increased risk for venous thromboembolism due to their underlying disease [40,41,42]. After the approval of the vector vaccines ChAdOx1 and Ad26.CoV2.S and application to millions of people, rare but significantly increased cases of cerebral thrombosis were detected especially in recipients of a younger age, while this was not the case after vaccination with mRNA vaccines [43,44]. This might provide another argument for the use of mRNA vaccines in IBD patients.

Although active IBD should not be a contraindication to vaccination in general, severe flares which require hospitalization and aggravated immunosuppression might require a postponed vaccine administration. If possible, vaccination should be performed while the patient is under stable therapy with the lowest possible level of immunosuppression. However, reducing immunosuppression just for the purpose of vaccine administration is not advisable, while waiting shortly for an already planned steroid taper, e.g., is rational. In any case, a thorough discussion and information of the patient based on individual circumstances is necessary [39].

In our clinical practice, we do not perform serology testing for SARS-CoV-2 before administering vaccination, even in people with suspected or proven prior infection, which is also in line with published recommendations [39]. It has been proposed that IBD patients should receive both doses of SARS-CoV-2 vaccination even if they have recovered from COVID-19, since data on the duration and strength of immunity after natural infections are missing [39].

Recently, booster immunizations have been proposed for selected patient cohorts and medical personnel and small studies in solid organ transplant recipients have suggested the application of a third dose of the BNT162b2 vaccine to enhance antiviral immunity [19]. At present, no studies with IBD patients exist to support this notion in the IBD population. It remains to be elucidated if prioritizing patients based on immunological profiles and clinical characteristics for a third vaccine dose could be useful. However, current national and international guidelines recommend booster immunizations six months after completion of the initial vaccine course, since protective immunity wanes over time, especially in elderly patients. Given the risk of suboptimal immune response in vaccinated patients under immunosuppression and the advent of new viral variants, booster immunizations should be considered for IBD patients, especially if the initial vaccination was performed under aggravated immunosuppression (which has possibly even been terminated meanwhile).

Our own meta-analysis of six available studies revealed an outstanding effectiveness of vaccination in IBD patients with a seroconversion rate of 96.4% in overall 676 participants up to 90 days after second vaccination. Nonetheless, the low number of available studies investigating the effectiveness and safety of SARS-CoV-2 vaccination in IBD patients and the small study size of these available studies are a relevant limitation in this review. Furthermore, not all studies differentiated in detail the IBD medication subgroups and the used vaccines in reporting the seroconversion rates, so that a meta-analysis on subgroups was not possible. A further limitation regards the influence of antibody concentrations on the effectiveness against severe disease in immunocompromised IBD patients. The available studies reported only in part absolute antibody concentrations; a meta-analysis was not possible due to different units in reporting the antibody concentrations. Larger studies are needed to investigate exact differences of immune responses and safety in IBD subgroups. Furthermore, available studies provide insufficient data regarding the influence of age, length of IBD history, type of IBD (Crohn’s disease vs. ulcerative colitis), and extraintestinal manifestations of vaccine response. However, the current literature points out that vaccinations against SARS-CoV-2 infections are safe and all in all effective in IBD patients and can strongly be recommended.

## Figures and Tables

**Figure 1 life-11-01220-f001:**
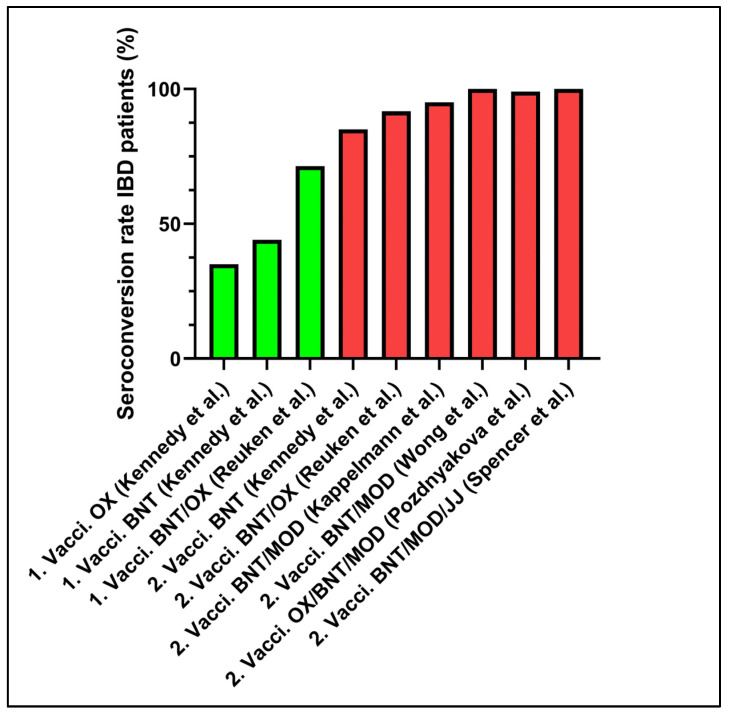
Seroconversion rates after the first and second SARS CoV-2 vaccination in IBD patients; 1. Vacci., first vaccination; 2. Vacci., second vaccination; OX, ChAdOx1; BNT, BNT162b2; MOD, mRNA-1273; JJ, Ad26.CoV2.S; IBD, inflammatory bowel disease.

**Figure 2 life-11-01220-f002:**
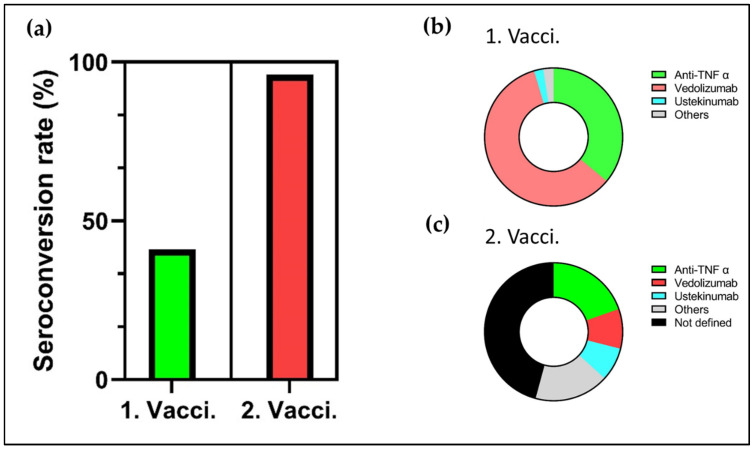
(**a**) Metanalysis of seroconversionrates after first vaccination (Kennedy et al., Reuken et al.; **n = 895**) and second vaccination (Kennedy et al., Reuken et al.; Kappelmann et al., Wong et al., Podzdyakova et al., Spencer et al.; **n = 667**). (**b**) Percentage of IBD patients on therapy with an anti-TNF α antibody, vedolizumab, ustekinumab, or other unspecified agents at the time of the first SARS-CoV-2 vaccination. (**c**) Percentage of IBD patients on therapy with an anti-TNF α antibody, vedolizumab, ustekinumab, or other unspecified agents at the time of the second SARS-CoV-2 vaccination. A significant number of therapies were not exactly defined or differentiated (black) by the original literature.

**Table 1 life-11-01220-t001:** Overview of available studies/literature and main results.

Study	Study Cohort	Vaccines	Results
Kennedy et al. [23]	IBD patients with: Infliximab (n = 865) Vedolizumab (n = 428)	BNT162b2 ChAdOx1	- Attenuated humoral response in infliximab compared to vedolizumab patients after both vaccines (1 dose) - Improved humoral response after 2 vaccine doses
Wong et al. [24]	IBD patients with: Anti-TNF-α (n = 16), Vedolizumab (n = 17) Vedolizumab/Thiopurin (n = 3) Ustekinumab (n = 4) oral Steroids (n = 3) No medication (n = 5) Healthy control subjects (n = 43)	mRNA-1273 BNT162b2	- No difference in seroconversion rates between IBD patients and healthy controls after full vaccination - No difference in seroconversion rates between IBD subgroups - Lower antibody levels in infliximab and vedolizumab patients compared to healthy controls - No difference in side effects between groups
Kappelmann et al. [25]	IBD patients with: Anti-TNF-α (n = 108) Anti-TNF-α/Combination (n = 24) 6MP/AZA/MTX (n = 20) 5-ASA/Budesonide/no medication (n = 65) Vedolizumab (n = 46) Ustekinumab (n = 39)	BNT162b2 mRNA-1273	- No difference in seroconversion rates between vaccination groups and IBD subgroups after full vaccination - Lower seroconversion rate in patients using steroids
Pozdnyakova et al. [26]	IBD patients with: Anti-TNF (n = 101) Anti-TNF-α/Combination (n = 32) 6MP/AZA/MTX (n = 20) 5-ASA/Budesonide/no medication (n = 65) Vedolizumab (n = 49) Ustekinumab (n = 94) Tofacitinib (n = 6)	BNT162b2 (2 doses, n = 193) mRNA-1273 (2 doses, n = 148) Ad26.CoV2.S (n = 12)	- Lower seroconversion rates and antibody levels after vaccination with Ad26.CoV2.S compared to BNT162b2/mRNA-1273 - No difference in seroconversion rates between IBD subgroups
Spencer et al. [27]	IBD patients with: Anti-TNF-α (n = 9) Ustekinumab (n = 9) Ustekinumab/Tofacitinib (n = 1) Tofacitinib (n = 1)	Ad26.CoV2.S mRNA-1273 BNT162b2	- A 100% seroconversion rate in children with IBD after full vaccination - No difference in antibody levels between IBD subgroups
Reuken et al. [28]	IBD patients with: Steroids (n = 2) Anti-TNF-α (n = 9) Vedolizumab (n = 3) Ustekinumab (n = 8) Azathioprin (n = 3) Mycophenolate (n = 2) Tacrolimus (n = 3) Tofacitinib (n = 1) Healthy controls (n = 27)	BNT162b2 ChAdOx1	- Significant cellular immune response in 27/28 immunocompromised IBD patients after 1 vaccine dose, similar to the response in healthy controls - Significant improvement of humoral response after second vaccine dose
Botwin et al. [29]	246 vaccinated IBD patients with different immunomodulating therapies	mRNA-1273 BNT162b2	- Rates of adverse events after 1 and 2 doses of vaccine in IBD patients comparable to rates known from vaccine registration trials

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
