# Peer review of "Vaccination against SARS-CoV-2 in Patients with Inflammatory Bowel Diseases: Where Do We Stand?"

_life, 2021, doi:10.3390/life11111220_

Round 1

Reviewer 1 Report

long expected review on current data about Covid-19 vaccination in IBD

there are sparse data and shortly upcoming data might change the landscape. Small number studies should be stressed to be of minor significance!

The ultimate goal will be determine the - severe- infection free intervals of vaccinated IBD patients depending on their therapies

Reviewer 2 Report

This review article is well written and is significant in present.

I think it is more beneficial to mention the comparison of other diseases to estimate whether IBD is a specific case or not. (Are there any remarkable diseases to care about the vaccination?)

Reviewer 3 Report

The authors present an interesting review regarding vaccination against SARS-CoV-2 in patients diagnosed with inflammatory bowel diseases. Although the idea of the paper is not bad, I believe it requires major revisions before publication. To my view, the paper would be more suitable as a short communication/letter to the editor rather than a full-text (mini)review.  

  1. The authors do not discuss the methodology employed to conduct this review. Considering the limited data available on the subject, I believe it would have been better to conduct a systematic review to ensure that all relevant publications were identified. In any case, the authors should discuss which databases were searched, which keywords were employed, the timeframe etc.
  2. Why did the authors choose to discuss only mRNA-based vaccines? There are countries in which other vaccines are currently administered. Please look for data coming from China, Russia etc.
  3. The paper requires a moderate polishing of the English language.
    • SARS-CoV-2 actually stands for severe acute respiratory syndrome coronavirus type 2 - of note, it is coronavirus not coronovirus. Please correct.
    • line 63 - folling - correct to following.
    • line 93 - check the spelling of infliximab etc
  4. Table 1 is not formatted according to the style required by the journal.
  5. The manuscript ends abruptly. Please add a separate section for Discussion in which you critically review the literature and provide recommendations regarding anti-COVID-19 vaccination in IBD. It is not sufficient to summarize the data from the literature without your own insight. Maybe it would also be interesting to briefly present some of your personal findings regarding this issue in the patients you manage in your center.
  6. The paper can also be improved by the addition of 1-2 figures and a graphical abstract.

Round 2

Reviewer 3 Report

Excellent work! Overall, the paper has been significantly improved and is now worth of publication in the journal. Just a minor correction - correct PubMed to PubMed/MEDLINE or to MEDLINE (PubMed is just the search service, not the actual database, i.e., MEDLINE).